# A Novel Method of Ship Detection under Cloud Interference for Optical Remote Sensing Images

**Wensheng Wang** [1,2,*], **Xinbo Zhang** [1], **Wu Sun** [3] **and Min Huang** [1,2]

1   School of Mechanical and Electrical Engineering, Beijing Information Science and Technology University, Beijing 100192, China
2   Key Laboratory of Modern Measurement & Control Technology, Ministry of Education, Beijing Information Science and Technology University, Beijing 100192, China
3   Artificial Intelligence Institute of China Electronics Technology Group Corporation, Beijing 100049, China
*   Correspondence: ws_wang@bistu.edu.cn; Tel.: +86-131-6176-9789

**Abstract:** In this paper, we propose a novel method developed for detecting incomplete ship targets under cloud interference and low-contrast ship targets in thin fog based on superpixel segmentation, and outline its application to optical remote sensing images. The detection of ship targets often requires the target to be complete, and the overall features of the ship are used for detection and recognition. When the ship target is obscured by clouds, or the contrast between the ship target and the sea-clutter background is low, there may be incomplete targets, which reduce the effectiveness of recognition. Here, we propose a new method combining constant false alarm rate (CFAR) and superpixel segmentation with feature points (SFCFAR) to solve the above problems. Our newly developed SFCFAR utilizes superpixel segmentation to divide large scenes into many small regions which include target regions and background regions. In remote sensing images, the target occupies a small proportion of pixels in the entire image. In our method, we use superpixel segmentation to divide remote sensing images into meaningful blocks. The target regions are identified using the characteristics of clusters of ship texture features and the texture differences between the target and background regions. This step not only detects the ship target quickly, but also detects ships with low contrast and under cloud cover. In optical remote sensing, ships at sea under thin clouds are not common in practice, and the sample size generated is relatively small, so this problem is not applicable to deep learning algorithms for training, while the SFCFAR algorithm does not require data training to complete the detection task. Experiments show that the proposed SFCFAR algorithm enhances the detection of obscured ship targets under clouds and low-contrast targets in thin fog, compared with traditional target detection methods and as deep learning algorithms, further complementing existing ship detection methods.

**Keywords:** superpixel segmentation; ship detection; cloud interference; optical remote sensing

## 1. Introduction

Ship detection has become a widely researched topic in optical remote sensing because of its applications in civil and military fields, such as ship rescue, fishery management and marine monitoring. In recent years, with the rapid development of optical satellite-based imaging technology, some researchers have devoted attention to ship detection with optical remote sensing images because of the good characteristics of high resolution and detailed spatial content of optical images. In the optical remote sensing image, ship detection tasks are disturbed by clouds, where sometimes thick clouds will obscure part of the ship hull causing only part of the ship to be visible, and no longer detectable when the whole ship detection method is used. At the same time, ships covered by thin clouds and ships with low contrast are only blurred and visible, which also affects the final detection results. These complex and special circumstances exacerbate the difficulty of detecting ship

targets in optical remote sensing images. The detection of ships at sea has been extensively studied [1–9], while less work has been carried out on the detection of ships under cloud interference. According to meteorological statistics, cloud coverage is usually around 40%, meaning that the effect of clouds cannot be ignored for sea-going ship testing. For such a task, we need to extract interesting ship targets from a great number of remote sensing images containing cloud interference. In recent years, many cloud detection algorithms and sea–land separation algorithms have emerged in the field of remote sensing images [10–14], which provide prerequisites for the detection of ships under clouds. The separation of sea and land can limit the detection area of ships and prevent the complex background on the land from affecting the target detection. Cloud region localization and cloud classification can set the scene for the application of our method.

This situation is more difficult than a clean background. When a ship is covered by clouds, ordinary ship detection methods are not applicable and may result in missing the target. Ships under clouds are difficult to detect by human eyes, which must look carefully to see them. Ships completely covered by thick clouds are also invisible to the human eye, so the algorithm cannot detect them and we ignore these cases. The gray information cannot be used because the grayscale is affected, and the ship does not show the initial state. The boundary also becomes blurred. Despite these obstacles, some details are still relatively obvious, namely, contours and textures, which make it possible to detect ships under thin clouds. The clouds in the image are usually homogeneous, but this homogeneity does not apply to ships under clouds.

Existing work [15–17] on ship detection under cloud interference can be divided into three categories. The first kind of clouds do not cover the target, i.e., the interference of small broken clouds. This kind of situation using the cloud shape and other information can be used to distinguished ships [18]. The second case is coverage by thin clouds. The processing method is to carry out thin cloud removal, which is also the most common method at present. However, this method loses the details of the original image. It can be used when there is no target under the cloud, but if there is a region of interest under the cloud, this method also loses the information of the target under the cloud [19]. Xu [20] of Peking University proposed an end-to-end feature fusion retention network (FFA-Net) to directly recover fog-free images, and the experimental results showed that the FFA-Net greatly surpassed the previous state-of-the-art single-image defogging methods in terms of quantity and quality, and improved the best peak signal to noise ratio (PSNR) on the SOTS indoor test dataset, from 30.23 to 36.39 dB. In the third case, some methods do not filter but directly detect the target, but these methods are for the last vessel, and clouds can be directly extracted by selecting a suitable gray threshold segmentation of the whole target, then analyzing the shape and other information of the extracted target to discriminate. This method is no longer applicable when the cloud's grayscale and the ship's grayscale values are close to each other, and when the coverage severity is relatively high. For example, Wang et al. [21] introduced an image defogging method in the target detection network to suppress the interference of clouds, and proposed a ship target detection method based on the SC-R-CNN network. A scene classification network (SCN) was used to achieve the classification of fog-containing images and cascaded with the target detection network to form a new target detection framework.

Deep learning-based target detection methods have been widely used in the field of optical remote sensing image target detection in recent years due to their automatic feature extraction and robustness [22–29]. Wang et al. [30] combined anomaly detection and a spatial pyramid pool, and proposed a ship target detection method based on SPP-PCANet. First, the images were segmented by land and sea, thus reducing a large amount of land interference information. Then, an anomaly detection algorithm was used to extract the candidate regions of the ship targets. Finally, the detection rate was further improved by SPP-PCANet. The authors tested the method on the images taken by GF-1 and GF-2 satellites, and the experimental results showed that the method had a good ship detection rate and low false alarm rate, with good robustness with background interference such

as low-contrast ships and uneven illumination. However, the overall method had high computational complexity and a long detection time. Moreover, deep learning requires a large number of datasets to be produced in advance, which is not easy for individual training because of the difficulty in obtaining the target data for this scenario and the variety of situations.

The ship scenes have similarities to SAR images when disturbed by clouds, that is, the background of the cloud environment around the target is relatively homogeneous. At the same time, while retaining the rich texture of optical remote sensing images, we thought of combining the two to design a method more suitable for our scenario [31–33]. We propose a new CFAR algorithm based on superpixel segmentation with feature points to solve the above problem. Our newly developed SFCFAR uses superpixel segmentation to divide the large scene into many small regions, which include both target and background regions. Superpixel segmentation serves as a preprocessing procedure to divide the image into meaningful patches. The target regions are then identified using the characteristics of clusters of ship texture features and the texture differences between the target and background regions. This method not only detects the ship target quickly, but also detects ships with low contrast and cloud cover. In optical remote sensing, ships under thin clouds in the sea are not common in practice, and the sample size generated is relatively small, so this problem does not apply to deep learning algorithms for training, while the SFCFAR algorithm does not require data training to complete the detection task.

In this research, we utilized a different idea of maritime target detection compared with previous methods. We did not detect clouds or fog, nor did we filter to affect the target, we only used the original data to detect the remaining features of the ship after being disturbed by clouds. Our proposed method is competent for three cloud-disturbed situations. In the first case, the target is disturbed by small broken clouds, but the clouds do not cover the target; in the second case, the target is partially obscured by clouds, and only part of the hull of the target can be seen; in the third case, the entire target is covered by thin clouds or mist, resulting in low contrast between the target and the background, but the target can be seen by careful observation. The remainder of the article is organized as follows: In Section 2, the theoretical part of our SFCFAR algorithm is presented. In Section 3, the experimental results of experiments in three parts are given. Section 4 discusses the results of the experiment, by comparison with other algorithms, from both qualitative and quantitative aspects, to demonstrate the advantages of our algorithm. Section 5 presents the conclusions.

## 2. Method Theory Explanation

The overview of our method is shown in Figure 1. Sea–land separation is not required because the Global Self-Consistent Hierarchical High-resolution Geography (GSHHG) database contains land and ocean boundaries for automatic land masking [34]. The GSHHG database was developed by the National Oceanic and Atmospheric Administration's Earth Science Laboratory and is openly accessible [35]. GSHHG geographic data provide coordinates of water bodies such as world coastlines, major rivers and lakes, and also five different resolutions: coarse, low, medium, high and full. In our task, full-resolution data were used to generate land masks. Our detection process was thus only carried out for seawater and was not disturbed by the terrestrial background.

The SFCFAR method includes three steps, i.e., hypothesis generation, candidate extraction, and target confirmation. In hypothesis generation, we employed superpixel segmentation to segment the image into several superpixels, and targets were contained in superpixel blocks. Some objects may be divided into multiple superpixel blocks. In candidate extraction, we used feature point extraction to locate candidate target positions to obtain target superpixel blocks. In this step, we also merged superpixels where an object was segmented into blocks. In target confirmation, we used superpixel-based CFAR to screen the target to obtain the final target.

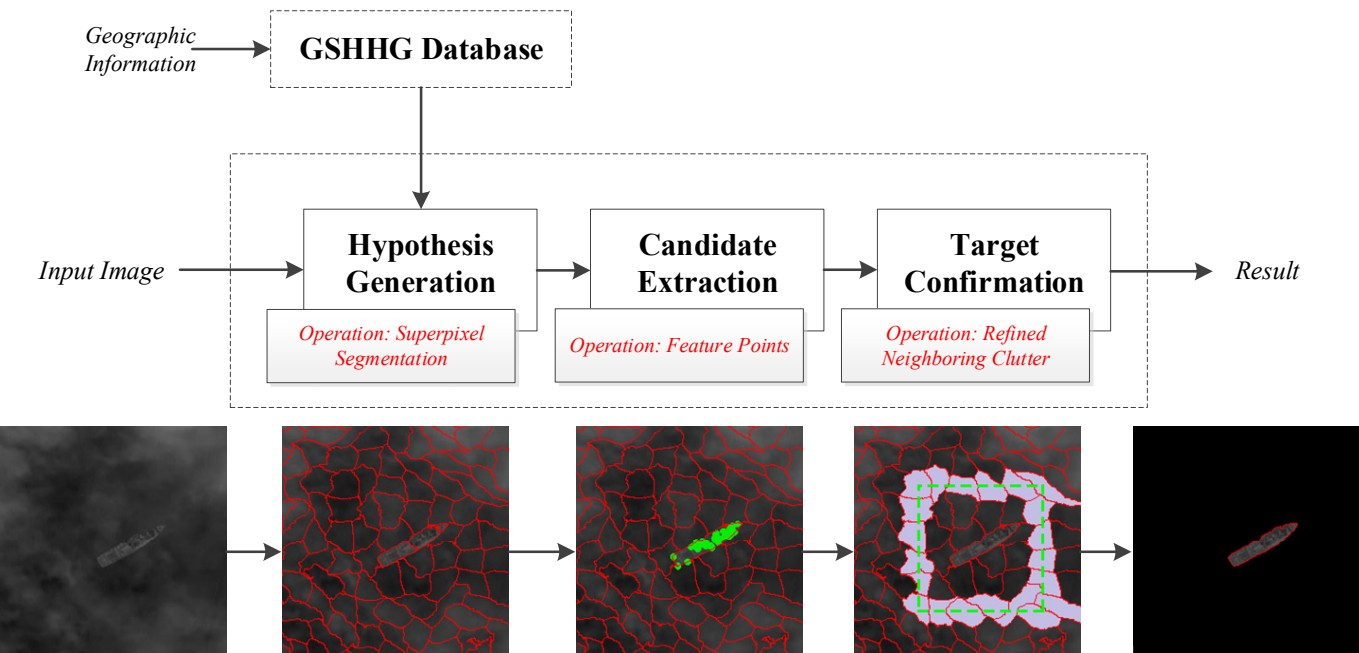

**Figure 1.** Ship detection procedure of the proposed method. First, we performed image masking with ocean regions as study areas, according to the GSHHG Database. Next, within the mask area, we performed hypothesis generation to perform superpixel segmentation of the image. Then, we performed candidate extraction. By extracting feature points, candidate regions were filtered and superpixels were merged. Finally, we confirmed the target through refined neighboring clutter and obtained the result.

Our experimental object was GF-1 optical remote sensing satellite data, the spatial resolution of panchromatic images was 2 m, the imaging spectral range was 0.45 to 0.9 μm; the spatial resolution of multispectral images was 8 m, and the spectral ranges of the four imaging bands were 0.45–0.52, 0.52–0.59, 0.63–0.69 and 0.77–0.89 μm.

### 2.1. Hypothesis Generation

For an image $I$ of pixel resolution size $N = W \cdot H$, the image width is $W$ and the height is $H$. The vector $\mathbf{z}_i = (x_i, y_i, c_i)^{\mathrm{T}}$ is used to declare the pixel $i$, where $(x_i, y_i)$ is the coordinate of the pixel $i$ and $c_i$ represents the intensity value of the pixel $i$. The superpixel algorithm requires as an input the required number of superpixels $K$, each corresponding to a Gaussian distribution. The Gaussian function $p$ is defined as in Equation (1), where the Gaussian parameters in the vector $\hat{\mathbf{z}}$ and the set $\hat{\theta} \stackrel{\text{def}}{=} \{\hat{\mu}, \hat{\Sigma}\}$ are separated by semicolons [36].

$$p(\hat{\mathbf{z}}; \hat{\theta}) = \frac{1}{(2\pi)^{\frac{D}{2}}\sqrt{\det(\hat{\Sigma})}} \exp\left\{ -\frac{1}{2}\hat{\mathbf{z}} - \hat{\mu})^T \hat{\Sigma}^{-1}(\hat{\mathbf{z}} - \hat{\mu}) \right\} \tag{1}$$

where $D$ is the number of components in $\hat{\mathbf{z}}$. Each superpixel $k \in N \stackrel{def}{=} \{0, 1, \cdots, K-1\}$ is a Gaussian distribution. Similar to $\hat{\theta}$, the parameter set $\theta_k$ is comprised of the mean vector $\hat{\mu}$ and the covariance matrix $\hat{\Sigma}$ corresponding to $\mu_k$ and $\Sigma_k$, respectively.

We assume that $\widetilde{L}_i \in K_i$ is a random variable of the superpixel corresponding to the pixel $i$. Then the observed value of the pixel $i$ at this time is the vector $\mathbf{z}_i$. The pixel-dependent Gaussian mixture model is defined as:

$$p_i(\mathbf{z}) = \sum_{k \in K_i} \Pr(\widetilde{L}_i = k) p(\mathbf{z}; \theta_k), \forall i \in V \tag{2}$$

where $\Pr(\widetilde{L}_i = k)$ is the probability that $\widetilde{L}_i$ takes label $k$, defined as the $P_i = 1/|K_i|$. Using $P_i$, Equation (2) simplifies to:

$$p_i(\mathbf{z}) = P_i \sum_{k \in K_i} p(\mathbf{z}; \theta_k) \tag{3}$$

This shows that the probability of obtaining pixel $i$ is the same for different Gaussian distributions. Therefore, the expected value of the superpixel is:

$$P_i |I_k| = \frac{1}{|K_i|} |I_k| = \frac{1}{9}(3v_x) \cdot (3v_y) = v_x \cdot v_y \tag{4}$$

That is, each superpixel is transformed to the equal size $v_x \cdot v_y$.

Once we have an estimate of the parameters in the set $\theta = \{\theta_k | k \in N\}$, we can obtain the label $L_i$ for pixel $i$ according to:

$$L_i = \arg_k \max_{k \in K_i} \Pr(Li = k | \mathbf{Z}_i = \mathbf{z}_i) \tag{5}$$

$$L_i = \arg_k \max_{k \in K_i} \frac{p(\mathbf{z}_i; \theta_k)}{\sum_{k \in Ki} p(\mathbf{z}_i; \theta_k)} \tag{6}$$

We hope that superpixels can evenly divide the image into small blocks; therefore, when the mean vector is initialized, it needs to use fixed-size $v_x$, $v_y$, and $K$ pixels, i.e., $\mu_k = \mathbf{z}_j$, where:

$$j = k_x \cdot v_x + \lfloor v_x/2 \rfloor + W \cdot (k_y \cdot v_y + \lfloor v_y/2 \rfloor) \tag{7}$$

The expectation is to initialize the covariance matrix near its final estimate, which is a local optimum representing similarity in size and homogenizing colors for fast convergence. For this reason, the spatial covariance matrix is initialized $\text{diag}(v_x{}^2, v_y{}^2)$. $\Sigma_{k,c}$ is initialized with $\text{diag}(\lambda^2, \lambda^2, \lambda^2)$, where $\lambda$ requires similarity between two pixels, so takes a relatively small intensity value. The square of $\lambda$ is used to initialize the main diagonal of the color covariance matrix. A small color distance is assigned to the parameter $\lambda$ so that the proposed method can achieve satisfactory accuracy within limited iterations. We assigned $\lambda$ as 2, 4, 6, 8 and 10, five different color distances, to show its effect on accuracy. In this study, we experimentally chose $\lambda = 8$, because it provides slightly better boundary recall than $\lambda = 10$ and generates fewer superpixels than other smaller $\lambda$ values.

Once the parameter set $\theta$ is initialized, it starts iterating until the tolerance is reached. In general, with this iteration stopping condition, the Gaussian distribution with more iterations is more pixel-friendly than with fewer iterations, which implies higher accuracy. Conversely, more iterations increase the complexity and efficiency of the overall operation. Hence, in the experiment, we set the number of iterations $T$ to 10, to balance the speed and accuracy of the operation.

Detecting the difference between the central pixels of the superpixels can suppress the influence of ocean waves or noise generated in the image imaging process [37]. We believe that normal pixels follow the Gamma distribution, and the difference between different superpixels can be expressed as follows:

$$\delta(u_1, u_2) = 2M \cdot L \cdot \ln \frac{\bar{I}_{u_1} + \bar{I}_{u_2}}{2\sqrt{\bar{I}_{u_1} \cdot \bar{I}_{u_2}}} \tag{8}$$

where $\bar{I}_{u_k}$ represents the average value in the superpixels $u_1$ or $u_2$, L is the number of image lookups, and $M$ is the number of pixels contained in the superpixel piece. In practice, for simplicity, $L$ in Equation (8) is taken as 1. Furthermore, choosing a block of $5 \times 5$ gives $M = 25$. The phase anisotropy metric is defined as follows:

$$D(i, j) = \delta(u_1 + u_2) + \lambda \cdot d(i, j) \tag{9}$$

where $d(i,j) = ((x_i - x_j)^2 + (y_i - y_j)^2)^{\frac{1}{2}}$ represents Euclidean distance, $(x,y)$ is the horizontal and vertical coordinates of the pixel $i$ and $j$; and $\lambda$ measures the significance of value differences $\delta(u_1, u_2)$ and spatial differences $d(i,j)$.

## 2.2. Candidate Extraction

The feature of thin clouds, which link up into a single stretch, is inconspicuous in this respect. If it can be detected by the human eye, a superpixel feature containing a vessel is stronger than no vessel. During candidate object extraction, potential object locations in the image were detected using Harris feature point, a local feature-based algorithm, proposed in [38]. However, the Harris point detection needs a threshold. We propose a self-adaptation method to automatically obtain the threshold for Harris point detection. The location of the salient points was, therefore, selected as a suspected target area location.

The Harris detector is a classic algorithm for feature point detection using the first derivative. It determines whether it is a feature point by judging the degree of change of the autocorrelation function in the horizontal and vertical directions of the image. The autocorrelation function is written as a matrix, and its eigenvectors are the main features of the detection points, which are defined as:

$$R = \det(C) - k \times \text{trace}^2(C) \tag{10}$$

where $k$ is an empirical scalar, generally taken as 0.04~0.06, and the matrix $C(x) = \begin{bmatrix} I_u^2(x) & I_{uv}(x) \\ I_{uv}(x) & I_u^2(x) \end{bmatrix}$. $I_u(x)$, $I_v(x)$ and $I_{uv}(x)$ are the partial derivative and second-order derivative of the gray intensity value of point $P$ in the $x$ and $y$ directions.

Since most of the remote sensing images in the ocean area are flat areas and have similar textures, there are few pixel points with prominent feature values. Thus, as the threshold value is calculated from large to small, there is a cliff change in the number of feature points, so we can automatically determine the threshold value in this image scene, and only select the feature point cliff threshold truncation. If the superpixel region is a non-target region, there are basically no feature points, or few feature points exist. In contrast, the hyperpixel of the target region contains multiple feature points and shows a clustering phenomenon. Generally, the target area of the superpixel is relatively small compared with the non-target area, and we defined the suspected target area by the percentage of target points to the number of pixels in the superpixel segmentation area. When the percentage of superpixel was more than 10%, it was considered as the superpixel of a suspected target area.

However, this resulted in a situation where a target was split into multiple superpixels, so we needed to merge the superpixels of the suspected target. After associating labels with pixels with Equation (6), we needed to enhance the connectivity between regions when the connectivity of each superpixel could not be guaranteed. We achieved region merging by merging a connected segment with its adjacent segments. Merging must commence with the smallest region. The color intensities of the two regions being merged were similar. First, we needed to determine the areas with the same label, and then sorted them by label number. Then, according to the order, it was estimated whether it needed to be merged. If the size of the current region was smaller than one quarter of the region size of the defined superpixel, then it was marked as a source segment. Among all the neighboring segments of the current source segment, superpixels with a similar number of feature points and matching the suspected target region were marked as target segments. Then the source segments were merged with their target segments. At the same time, the size of the source segment was merged into the size of the destination segment. When no further merging was possible, a new superpixel was formed. Before the post-processing step, it is theoretically guaranteed that superpixel $k$ will not appear in regions other than $I_k$.

*2.3. Target Confirmation*

For the target region confirmation work, i.e., for the superpixel under test (SUT) confirmation, we used the method in Figure 2 to obtain the background superpixel for clutter parameter estimation, i.e., the method of neighboring outward superpixels. After selecting the neighborhood superpixel centered on the SUT, the external superpixel of its neighborhood superpixel was determined, and the final background superpixel region was the nearest neighbor superpixel that did not border the target region superpixel. In Figure 2, the lavender color area is the neighboring superpixel of the SUT, and the purple area is the final background superpixel.

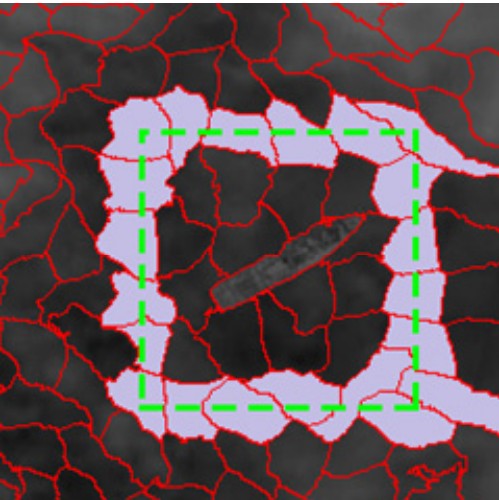

**Figure 2.** Reference window setup. The tested superpixel is the superpixel in the center of the green dashed box, and the surrounding clutter superpixels (shown in lavender) are selected for clutter parameter estimation. The selected lavender superpixel is the nearest non-adjacent superpixel of the tested superpixel.

Target confirmation also required the calculation of the estimated target eigenvalues within the superpixels, and we used the method of Li [39] to calculate the truncated clutter statistics and estimate the thresholds for target confirmation. For the $m$th SUT, at the specified false alarm rate $P_{fa}$, the threshold $T_m$ is as follows:

$$1 - P_{fa} = \gamma(\frac{\hat{L}_m T_m}{\hat{\mu}_m}, \hat{L}_m)/\Gamma(\hat{L}_m) \tag{11}$$

where $\hat{L}_m$ and $\hat{\mu}_m$ are estimated parameters for truncated Gamma distributions with background superpixels. The pixels included in the SUT are determined according to the following judging criteria:

$$\begin{cases} I_m(k) \geq T_m, & \text{if the pixel belongs to the target area} \\ I_m(k) < T_m, & \text{if the pixel belongs to the clutter area} \end{cases} \tag{12}$$

where $I_m(k)$ represents the intensity of the $k$th pixel in the $m$th SUT, and the number of $k$ in each SUT is different. If $I_m(k)$ is greater than the threshold $T_m$, the $k$th pixel in the detection result is set to 1; otherwise, the $k$th pixel is set to 0. After all pixels are detected, we can obtain a binary image.

The histogram of the clutter region contaminated by outliers had excessive peaks and upward tails. In the histogram, the regions with lower statistical values were generally the pixels of the background region, and the regions with higher statistical values were generally the pixels of the target region. However, there were many outliers. We reduced the influence of these abnormal pixels by setting the truncation depth. Supposing that the histogram of an image was set to k gray levels, we defined peaks at least 2K/3K bins

apart as the boundary between background and target, and defined the truncation depth as $t = (n_1 + n_2)/2K$, where $n_1$ and $n_2$ represented the corresponding positions of the two peaks, respectively.

## 3. Experimental Results

In this section, we show the whole experimental process and the experimental results using several pictures, zooming in on the more typical targets. As in Section 2, we present the experimental results of each step in three stages: hypothesis generation, candidate extraction, and target confirmation. Figure 3 shows the color image tested. Figure 4 shows the panchromatic images and the detail images tested. These images were all from the PMS sensor of the GF-1 satellite, with a resolution of 2 m panchromatic/8 m multispectral. Their specific latitude, longitude and date of photography are shown in Table 1. From the figure, we can see that the target in the color image is more difficult to find, and the target can be seen in the full-color image on close inspection. There are two reasons for this phenomenon. First, the color image was obtained by a multispectral sensor, and its resolution was lower than that of the panchromatic image, which led to fewer target pixels and affected the observation. Second, in the color image, because the background cloud was white, it was easier to cover the target, which was not conducive to observation. For example, several objects in the upper left corner of Figure 3c can vaguely be seen in the full-color image, but are difficult to observe in the color image.

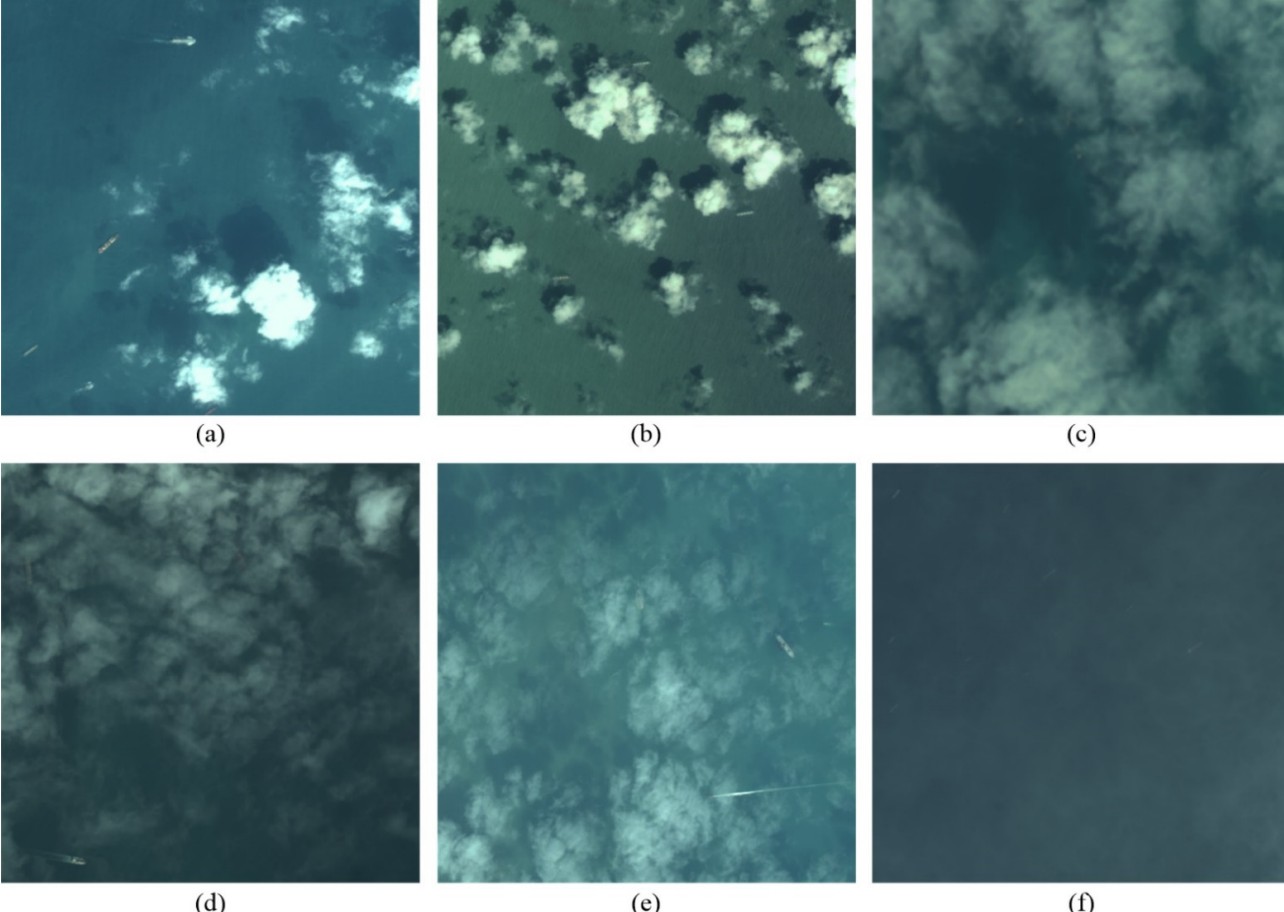

**Figure 3.** Color image of GF-1 disturbed by clouds to be detected. (**a**,**b**) are disturbances from broken clouds, where part of the target's hull is obscured. (**c**–**e**) are thin cloud coverage, in which part of the target is completely covered, but the outline of the entire hull is still recognizable. (**f**) is a low-contrast image under the influence of fog, and the gray values of the target and background are relatively close.

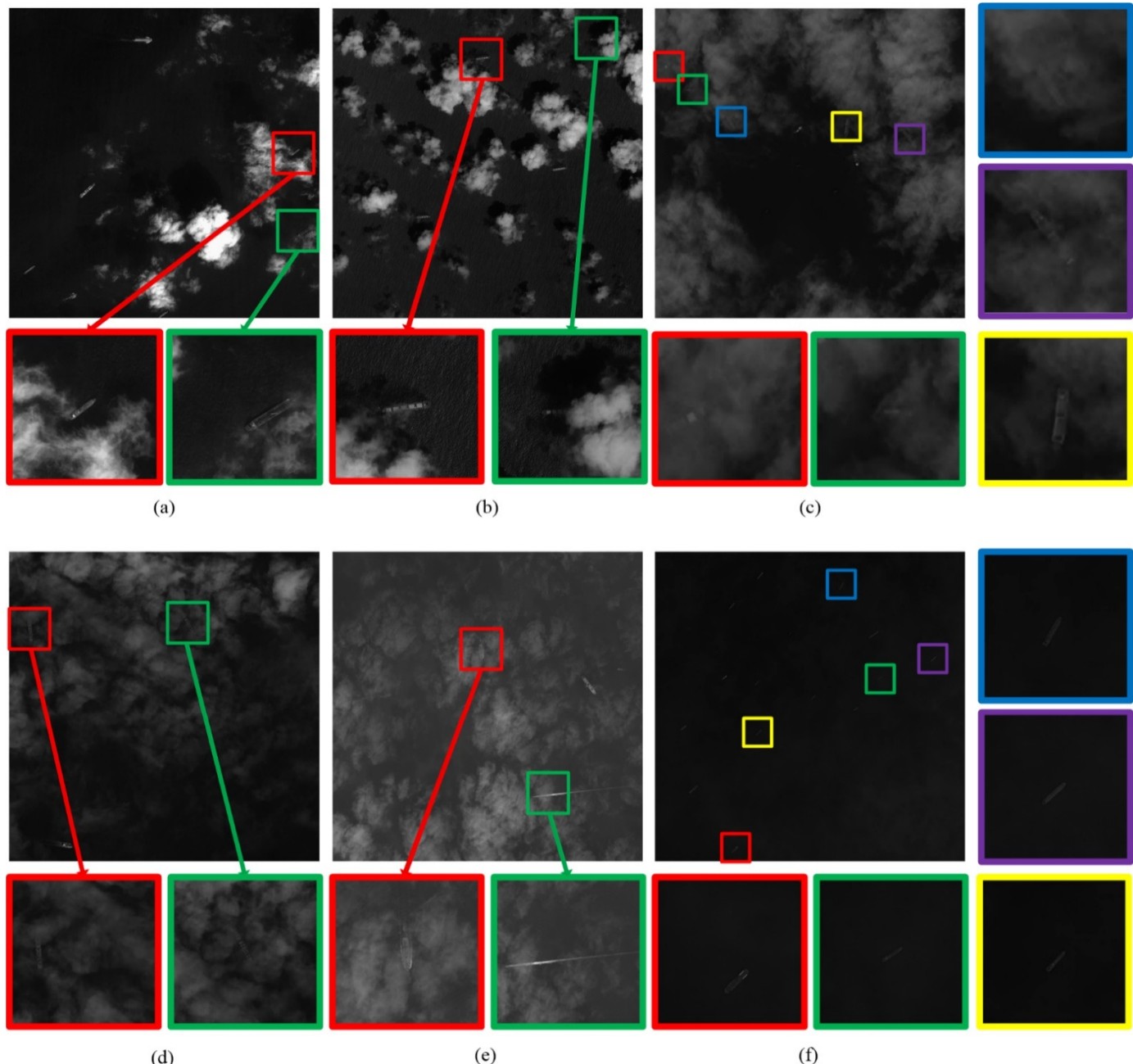

**Figure 4.** The panchromatic image corresponds to the color image in Figure 3 and the enlarged view of the details. Colored boxes are typical target areas; well-recognized targets are not marked and magnified. The target in (**a**) is connected to the cloud. The target framed in green in (**b**) is half of the hull occluded by the cloud. (**c**–**e**) are thin cloud coverage, in which part of the target is completely covered, but the outline of the entire hull is still recognizable. (**f**) is a low-contrast image, and the gray values of the target and background are relatively close.

**Table 1.** The information corresponding to each image in Figure 3. It includes the latitude and longitude information, and imaging date information, of each subfigure in Figure 3. The calibration type of the image is radiance.

| Number | Latitude and Longitude | Date of Photography | Calibration Type |
|:---:|:---:|:---:|:---:|
| (a) | E114.1_N22.1 | 31 October 2016 | |
| (b) | E122.3_N31.4 | 13 August 2016 | |
| (c) | E113.8_N22.4 | 24 December 2015 | radiance |
| (d) | E114.5_N22.4 | 8 December 2014 | |
| (e) | E113.7_N22.4 | 15 December 2016 | |
| (f) | E121.8_N38.9 | 31 August 2015 | |

### 3.1. Hypothesis Generation Results

Figure 5 illustrates the experimental results of target candidate region extraction. We can clearly see from the figure that most of the targets, whether covered by thin clouds or in low contrast conditions, can be super-resolved to separate the targets, but some targets were divided into two or three superpixel regions, which were integrated into the later steps. Moreover, when the background was relatively homogeneous, the shape of the superpixel segmentation was basically square. When the target existed, the contour of the superpixel segmentation was basically in agreement with the contour envelope of the target. In particular, those typical targets that were not easily found by clouds, as marked in Figure 4, could be segmented in this step. It can be seen that the purpose of extracting target candidate regions by superpixel segmentation is achieved.

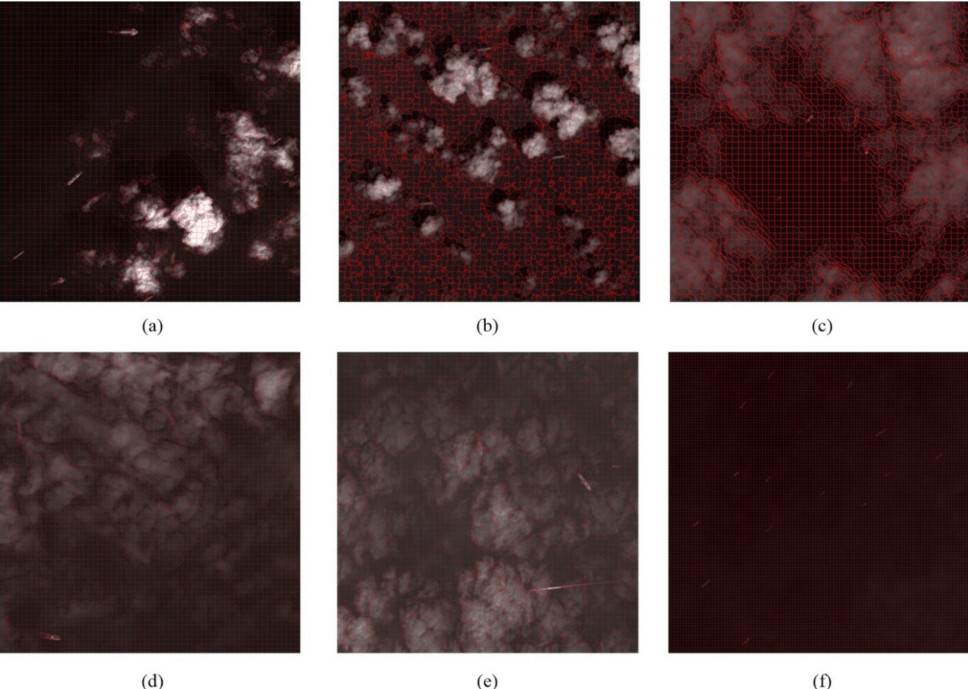

**Figure 5.** The superpixel segmentation images using the Figure 3 images. (**a**–**f**) are the superpixel segmentation images using Figure 3a–f. The target area is segmented by superpixels, and some of the targets are segmented into multiple superpixels. The superpixel block is segmented by the red line. We can see that in a uniform background area, the superpixel blocks are square; in areas with targets or disturbed by clouds, the edges of the superpixel blocks are irregular.

### 3.2. Candidate Extraction Results

In our hypothesis generation, as described in Section 2.2, we did not need to use our improved algorithm to select the thresholds for Harris feature point detection. Figure 6 shows the experimental results of target candidate region extraction. It can be seen from the

figure that the feature points we used could hit the target area well, regardless of the thin cloud-covered vessel or the bare leakage of the vessel semi-obscured by thick clouds. At the same time, due to the cluster effect of feature points, there were many and concentrated feature points in the target area. After the operation of this process, we could basically target the candidate region superpixels of the target, and at the same time, merge the cut-off target regions effectively. Figure 7 is an enlarged view of Figure 6e.

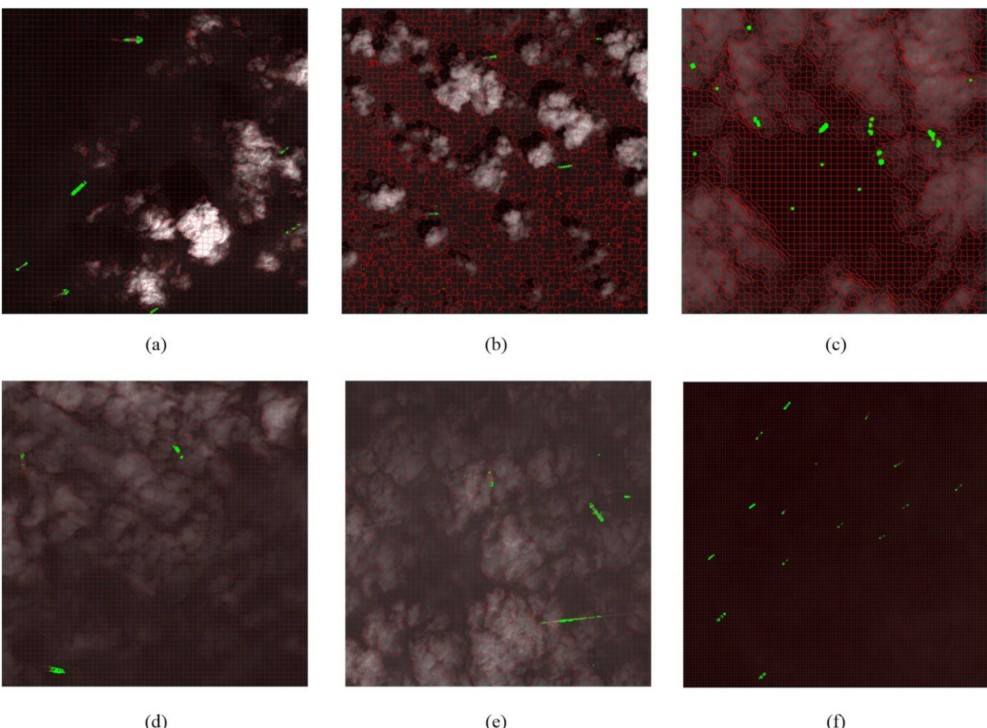

**Figure 6.** Feature point detection image. (**a–f**) are the feature points images using Figure 5a–f. Combined with superpixel segmentation and feature point detection, the divided superpixel target areas are merged into multiple blocks, and then judged as to whether it is a suspected target.

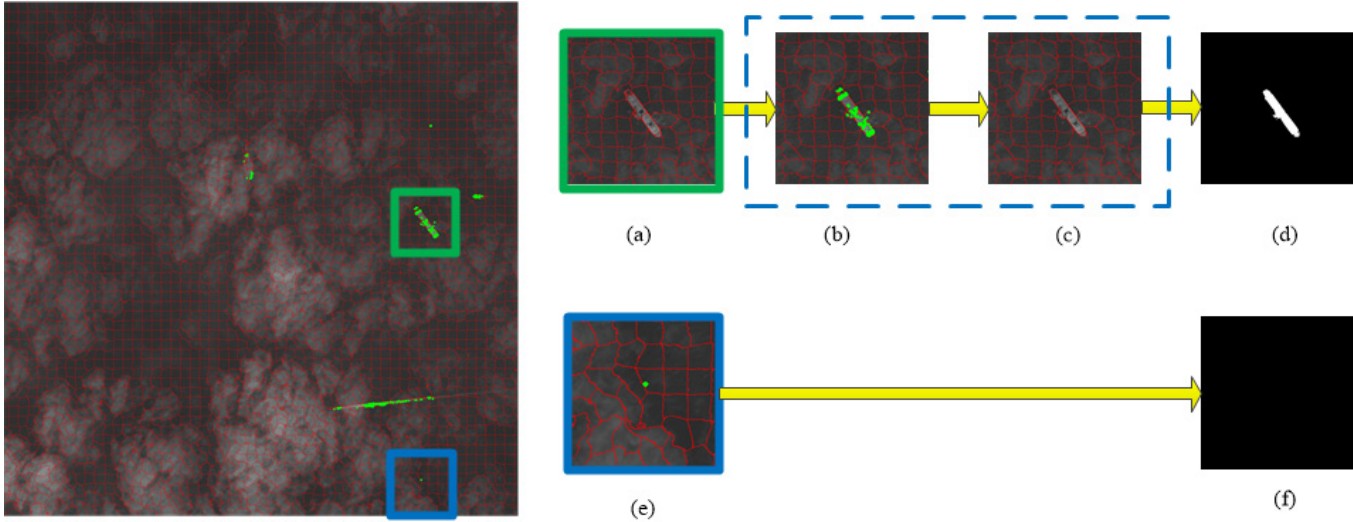

**Figure 7.** Taking (**e**) in Figure 5 as an example, the process of merging the target superpixel region (green box) and the culling process of non-target feature points (blue box) is shown. (**a–d**) are the results of feature point detection and superpixel merging in the target area. (**e,f**) are the culling results of non-target area feature points.

### 3.3. Target Confirmation Results

In the confirmation of the suspected target superpixel (SUT), we used the neighborhood external superpixel method to obtain background superpixels for clutter parameter estimation on the results of the previous step, and finally obtained the target region, as shown in Figure 8.

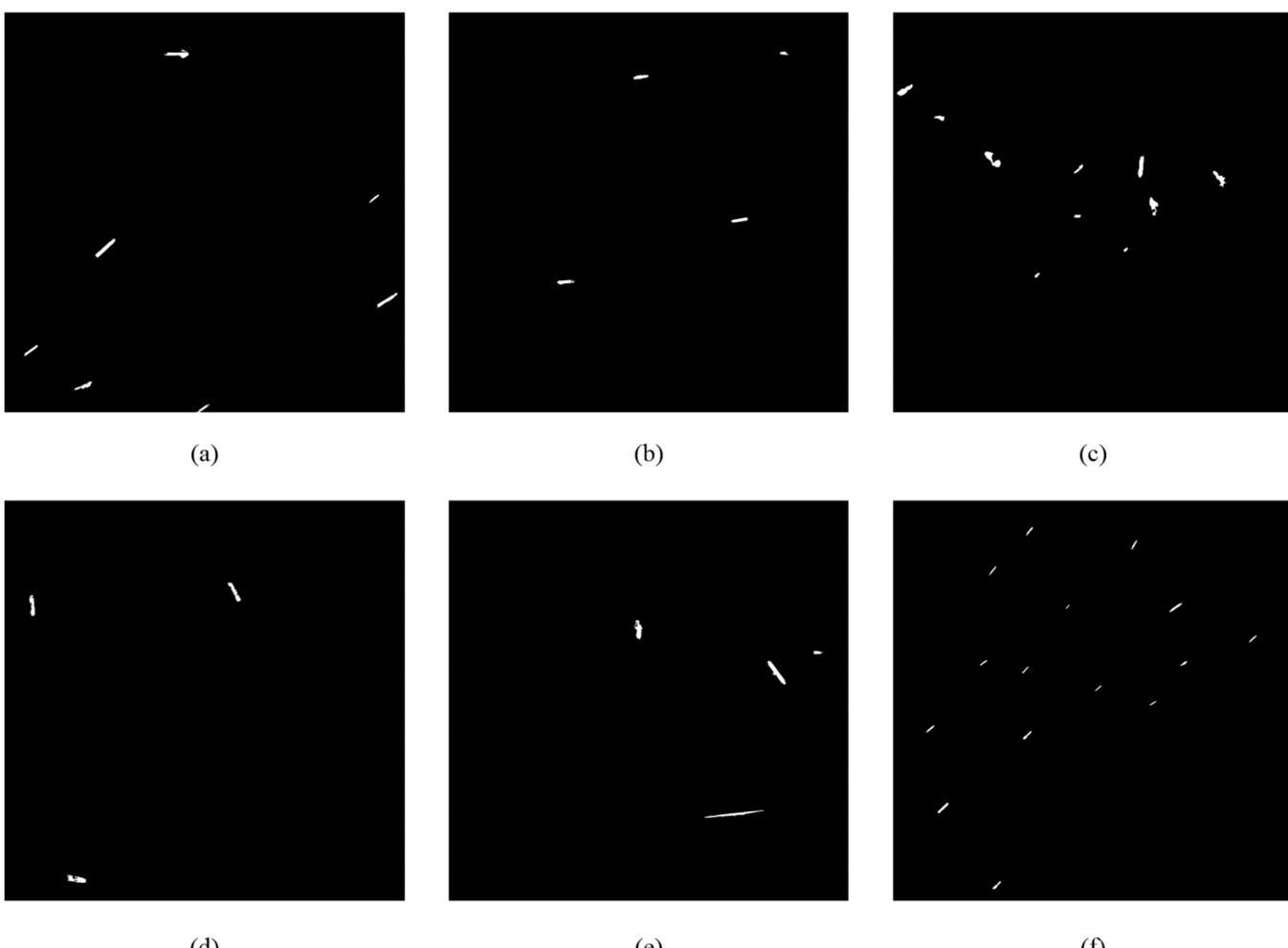

(a) (b) (c)

(d) (e) (f)

**Figure 8.** Target detection segmentation result image. The target superpixels are set to white, and the background superpixels are set to black. (**a**–**f**) are the final target segmentation results in Figure 3a–f.

Figure 9 shows the final detection result of the example image in Figure 3. From the figure, we can see that ship targets that cannot easily be observed by the human eye in the color remote sensing image, were also detected by the SFCFAR algorithm. Especially in Figure 9c,d, some ships can only be seen traces of targets in the panchromatic map. This kind of target is difficult to find with existing methods, and our SFCFAR can use the feature points of the abnormal area very well to locate the target, thereby recognizing the existence of the target. Figure 10 shows some details of Figure 9, where the detection results can be seen more intuitively, proving the superiority of SFCFAR. Figure 9f shows some low-contrast images disturbed by fog; our algorithm could detect all objects easily.

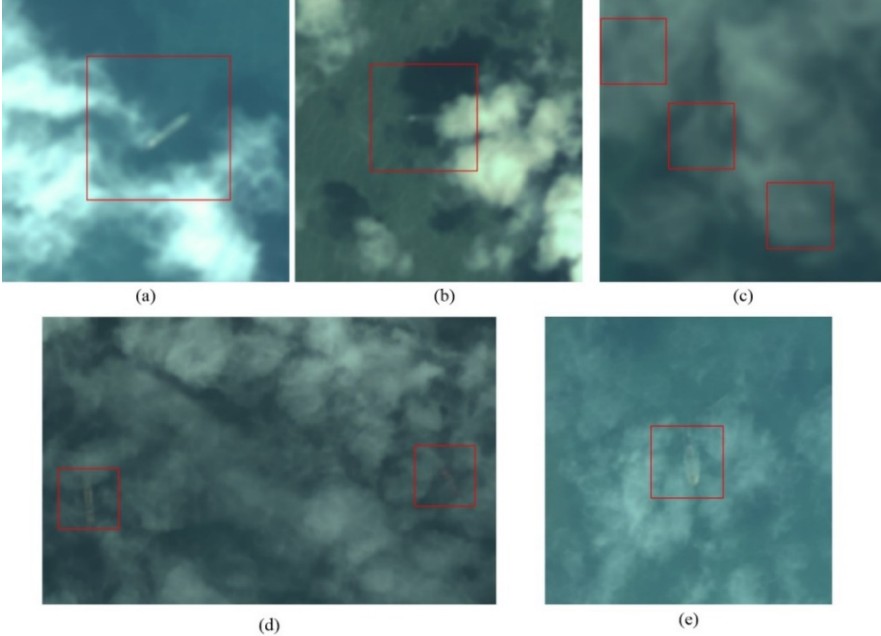

**Figure 9.** Final detection results of the images in Figure 3. The red square is the mark of the detected target. We can see that the objects in the graph are detected with our method. (**a–f**) are the final target detection results marked in red squares using the images of Figure 3a–f.

**Figure 10.** Detailed views of (**a–e**) of Figure 9. Enlarging the image can clearly see that the target blocked by the cloud has been detected.

## 4. Discussion

We conducted numerous experiments using visible panchromatic images and four-band multispectral images taken by the GF-1 PMS optical remote sensing satellite sensor, and evaluated our method quantitatively. The GF-1 PMS panchromatic band images had a spatial resolution of 2 m, and the four-band multispectral images had an 8 m spatial resolution. We selected a large number of images with cloud interference as the test set for testing, and we carefully checked the images of each scene to obtain the true value of the ship target. The true value was based on the target we could see with the naked eye in the panchromatic image (very weak targets were also counted).

These images contained various types of ships located under different kinds of clouds, but these ships were generally catchable by human eyes. Some parts of some of the ships were obscured; others were covered by thin clouds but could be seen as ships by the human eye. The image sizes of the test data ranged from 1000 × 1000 pixels to 10,000 × 10,000 pixels, and these images contained thousands of ships covered by various clouds. The proposed method was implemented in C++ with an Intel (R) Core (TM) i7-4770K CPU at 3.40 GHz and 64.0 GB RAM. The computer operating environment was Windows 10, and the running software was Visual Studio 2017. Due to the different test image sizes, the average calculation speed per 1000 × 1000 pixels was 39.6 ms.

We compared the SFCFAR algorithm with the more popular deep learning algorithms and traditional algorithms. Figure 11 shows the results of the comparison with the method proposed by Nie [15]. Table 2 shows the specific shooting information of the image in Figure 11. The first row in the figure is the data image of the GF-1 remote sensing satellite, the second row is the detection result of our SFCFAR algorithm, the third row is the detection result of the Nie algorithm, and the fourth row is a partial enlarged image. Red boxes represent detected ships, green circles represent undetected targets, and blue circles represent detected false targets. From Figure 11, we can see that when using Nie's method, objects occluded by the shallow cloud could be detected. However, when the degree of cloud occlusion was large, Nie's method failed. Moreover, our SFCFAR algorithm also outperformed Nie's method for incomplete objects and low-contrast targets.

In order to quantitatively analyze the stability of the method, we evaluated the precision and recall, which are most commonly used in the field of object detection, as performance metrics. Recall is the ratio of the number of correctly detected objects (true positives (*TP*)) to the total number of objects in the image (true positives (*TP*) and false negatives (*FN*)). Similarly, accuracy is the ratio of the number of correct targets found (true positives (*TP*)) to the total number of targets found that are believed to be true (true positives (*TP*) and false positives (*FP*)).

$$recall = \frac{TP}{TP + FN} \tag{13}$$

$$precision = \frac{TP}{TP + FP} \tag{14}$$

Figure 12 shows the PR curve for the comparison of the experimental results. Among them, the deep learning algorithm was verified by the most popular YOLO v5 algorithm [19,40]. Two compared deep learning methods were compared, one directly using a large-scale benchmark dataset "DIOR" proposed by Northwestern Polytechnical University for object detection in optical remote sensing images to train the model [7], which we called method 1. The dataset used in method 2 was a new dataset formed by adding our cloud interference data to the "DIOR" dataset. We named it "CDIOR". At the same time, we also compared the method proposed by Nie. In the results, the accuracy of our algorithm was significantly higher than other algorithms for targets obscured under clouds, for targets under thin clouds and low-contrast targets in cloudy conditions, and our algorithm also had obvious advantages. Methods 1 and 2 both used the popular deep learning algorithm YOLO v5. We chose the same method but different datasets for comparative experiments to show that deep learning is largely dependent on the establishment of datasets. As cloud occlusion conditions are too variable and complicated, datasets cannot cover all of them; in

some cases, it is also necessary to test factors such as deep learning sample enhancement. However, our algorithm did not need to rely on the establishment of the dataset to achieve good results, and it can be seen, from Figure 12, that it was better than the above methods.

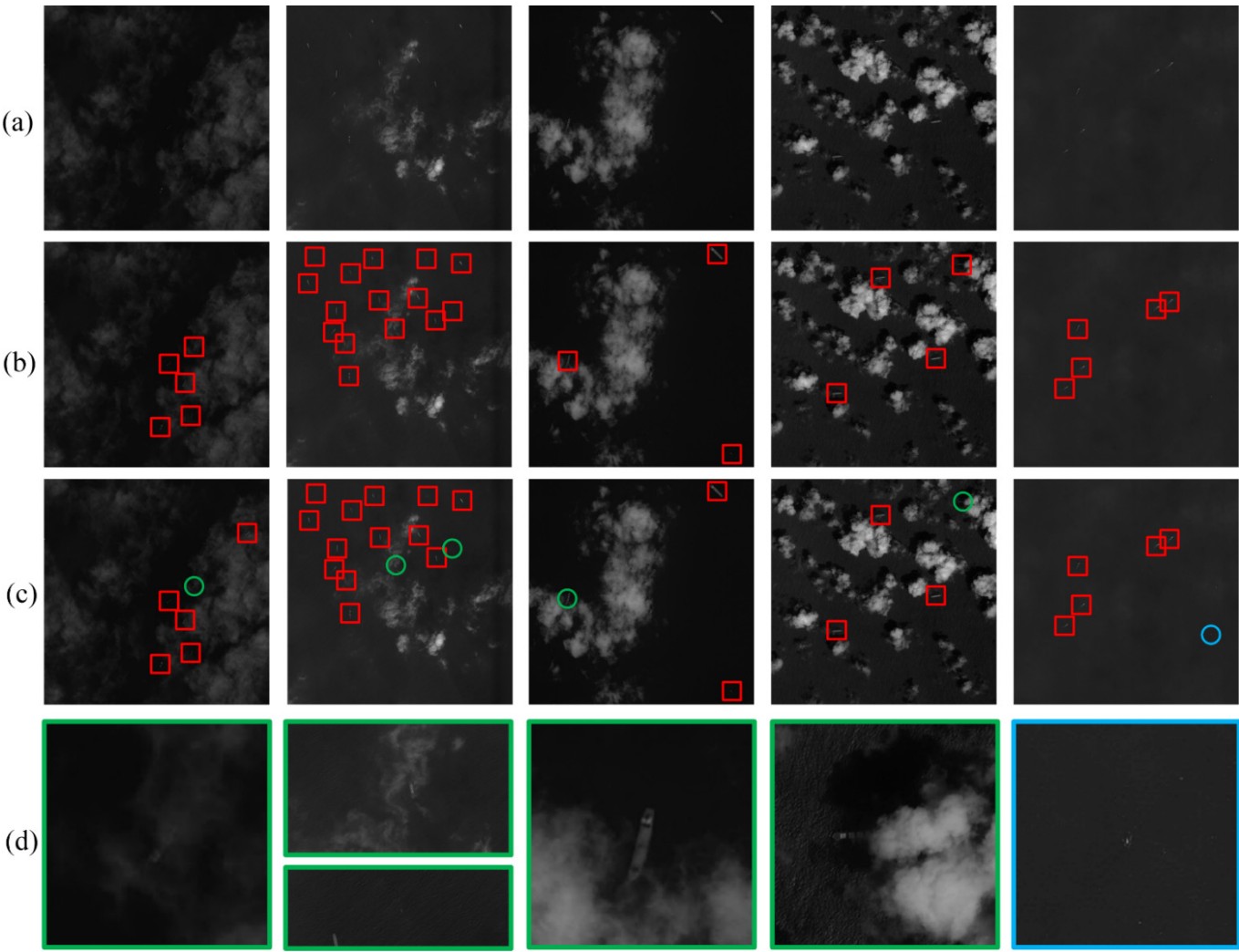

**Figure 11.** The experimental results are compared with images and detailed enlarged images. (**a**) the data image of the GF-1 remote sensing satellite; (**b**) the detection result of our SFCFAR algorithm; (**c**) the detection result of the Nie algorithm; (**d**) a partial enlarged image. Red boxes represent detected ships, green circles represent undetected targets and blue circles represent detected false targets.

**Table 2.** The information corresponding to each image in Figure 11. It includes the latitude and longitude information, and imaging date information, of each subfigure in Figure 11. The calibration type of the image is radiance.

| Number | Latitude and Longitude | Date of Photography | Calibration Type |
|:---:|:---:|:---:|:---:|
| 1 | E113.8_N22.4 | 24 Decmber 2015 | |
| 2 | E113.8_N22.2 | 31 October 2016 | |
| 3 | E114.2_N22.2 | 15 August 2014 | radiance |
| 4 | E122.3_N31.4 | 13 August 2016 | |
| 5 | E121.8_N38.9 | 9 November 2015 | |

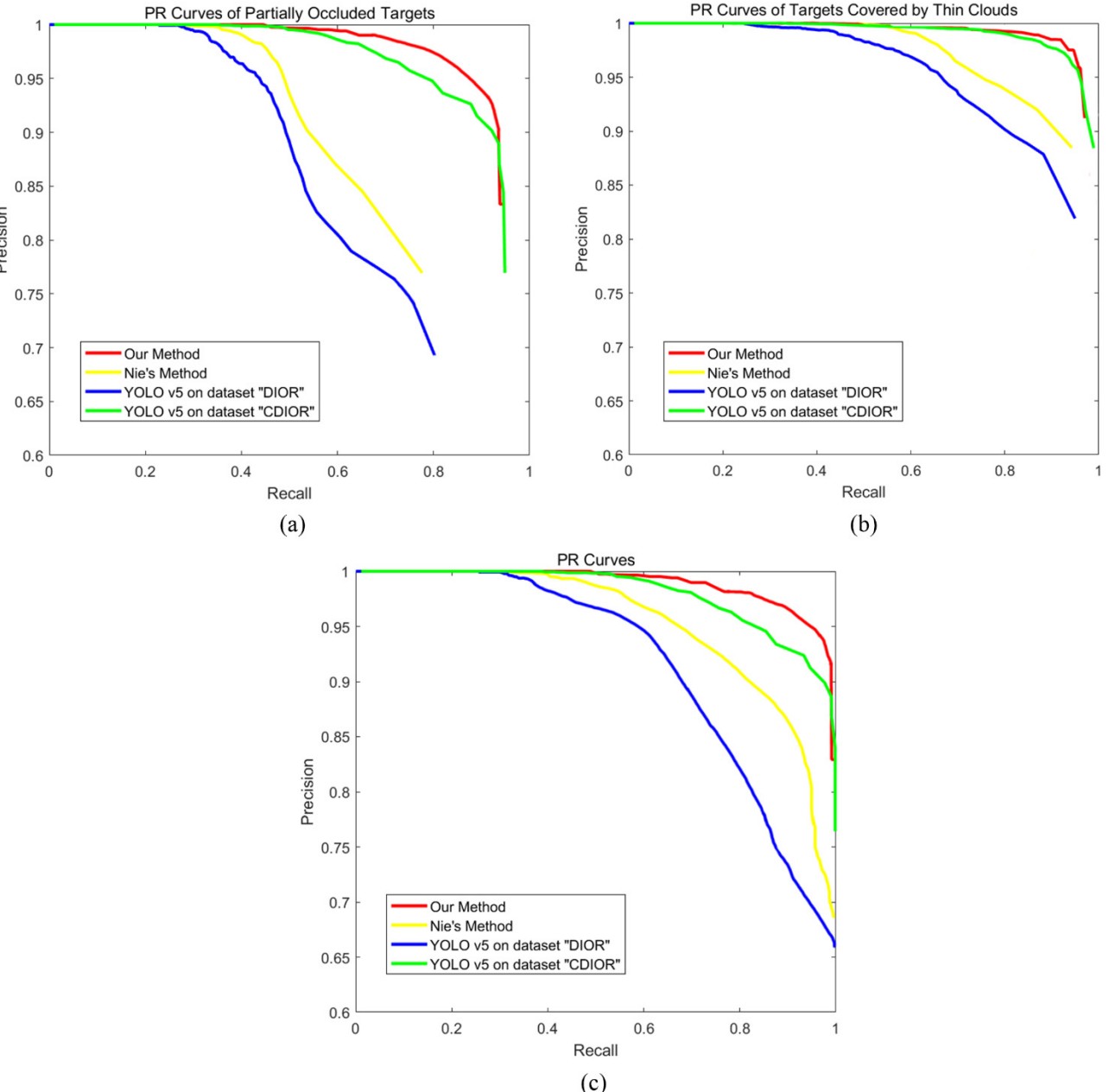

**Figure 12.** Comparison of PR curves of different methods: (**a**) the PR curve of the target partially occluded by thick clouds; (**b**) the PR curve of the target covered by thin clouds; (**c**) the PR curve of the overall result. We compared the method proposed by Nie [15], YOLO v5 on dataset "DIOR" and YOLO v5 on dataset "CDIOR".

Finally, we also introduced three indicators of *accuracy*, *missed alarm rate* (*MA*) and *false alarm rate* (*FA*) for comparison, where the *accuracy* is the *recall* in Equation (13). The evaluation criteria of the *MA* and *FA* are defined thus:

$$Accuracy = \frac{TP}{TP + FN} \tag{15}$$

$$MA = \frac{FN}{TP + FN} \tag{16}$$

$$FA = \frac{FP}{TP + FN} \tag{17}$$

As shown in Table 3, the experimental data we selected were the typical materials of various cloud shielding files. The effect of different methods on cloud interference is clear. The YOLO v5 deep learning algorithm (Method 1) with our cloud interference dataset was significantly better than the DIOR dataset alone, and also better than Nie's method. This shows that deep learning algorithms are very dependent on datasets. However, there are many kinds of cloud cover files, and the dataset is not easy to establish. Our method achieved an accuracy of 90.4%, which was significantly better than other methods. At the same time, its MA and FA were also better than those of other methods, which were controlled at around 10%. This means that it already achieves excellent results when using cloud cover to interfere with the experimental material. If some clean scene experimental material were to be added, the experimental result data will be improved.

**Table 3.** Comparison of experimental results of different methods. The comparison parameters used are accuracy, MA and FA. The comparison methods are YOLO v5 on dataset "DIOR", YOLO v5 on dataset "CDIOR", and Nie's Method.

| Method | *Accuracy* | *MA* | *FA* |
|---|---|---|---|
| YOLO v5 on dataset "DIOR" | 68.2% | 31.8% | 42.1% |
| YOLO v5 on dataset "CDIOR" | 81.9% | 18.1% | 28.5% |
| Nie's Method | 78.3% | 21.7% | 39.6% |
| Our method | 90.4% | 9.6% | 10.8% |

## 5. Conclusions

In this paper, we proposed a method for detecting cloud-interfering naval targets in optical remote sensing images. The target region of the image was segmented from the background region by super-resolution, at which time an initial region segmentation was obtained and the target region could be segmented into multiple parts. Then, the target super-resolved regions were merged using the characteristics of clusters of ship target feature points. Finally, the target region was confirmed using the constant false alarm rate detector of superpixels to obtain the target region. The method effectively overcomes the interference of clouds and has a good detection effect for ships with thick clouds obscuring part of the hull, ships under thin cloud coverage, and ships with low contrast. The method does not require a priori knowledge or training of the dataset to complete the detection task well, and it has good application value for target detection of such niche data. Moreover, it can be seen from the experimental results that the method has a high accuracy and is not inferior to the current mainstream deep learning algorithms.

**Author Contributions:** Conceptualization, W.W.; methodology, W.W.; software, W.W.; validation, W.W. and W.S.; formal analysis, W.W.; data curation, W.W.; writing—original draft preparation, W.W.; writing—review and editing, X.Z., W.S. and M.H.; project administration, M.H.; funding acquisition, M.H. All authors have contributed significantly and have participated sufficiently to take responsibility for this research. All authors have read and agreed to the published version of the manuscript.

**Funding:** This work was supported in part by the National Natural Science Foundation of China (Grant No. 61801455).

**Data Availability Statement:** Not applicable.

**Conflicts of Interest:** The authors declare that they have no conflict of interest.

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
