# Peer review of "A Novel Method of Ship Detection under Cloud Interference for Optical Remote Sensing Images"

_remotesensing, doi:10.3390/rs14153731_

Round 1
Reviewer 1 Report
The authors developed a new method of ship detection under cloud interference for optical remote sensing images, and performed extensive experiments using GF-1 data, with a comparison with other mature algorithms. The paper looks interesting and is in high demand, but requires a major revision before publication.
Detailed comments are below.
1. The English written of this paper is difficult to read and understand. A number of statements in this paper need to be revised. Here I only list few expressions in the first few pages, e.g. P1 L15, the expression “and study its application” should use passive tense to match the former. P1 L20, constant false alarm rate. P2 L56, which make the cloud region localization. I suggest the authors seek a native English speaker for help and double check every sentence. Otherwise, the paper is not suitable for publication.
2. P2 L50, the reference format [1-9] is not specific enough.
3. P3 L123, the authors seem to apply the idea or method of ship detection for SAR images to the proposed method for GF-1 images. Therefore, I don't think it's appropriate to say "we have a completely different approach to detecting objects at sea compared with previous methods".
4. P8 Figure3, these colorful remote sensing images from GF-1 should have the values of latitude, longitude, and radiance or reflectivity added.
5. P15 Table 1, the experiment results were compared with those of other algorithms to demonstrate the validity of the proposed method, using three parameters of accuracy, MA, and FA. The authors should state from which datasets or other sources the true value of ship detection method is, otherwise the TP and FN seem not credible. This is extremely critical for detection algorithms, especially for the cases under cloud interference.
6. P15 Figure 12, the expression in the legends is irregular.
Author Response
Please see the attachment. Thank you for your suggestion.

Reviewer 2 Report
In this paper, the Authors propose a method to detect ship within optical remote sensing imagery containing clouds. The proposed technique is a mixture of several machine learning&image processing methods (superpixels, constant false alarm rate....). That is, it is not an impressive paper and its main merit is that it is well presented and have a fine experimental setup. However, in this days where all turns around deep learning, this reviewer acknowledges the effort done by the Authors.
Concerns:
- Discussion section is too short. More has to be said in the comparison analysis. Figures shown (12) has to be discussed.
- Also, it is not clear to name "method 1" and "method 2" to a deep learning technique (the same?) but using different training set.
- It seems that the reference to Nie's paper is wrong. The Authors say "Nie 11" (many times), but it seems that it should be Nie 15. Revise it.
- In Table 1, Method form [148]--> revise (there is no such reference 148 in the references list).
- Computational cost and implementation details must be included (and compared with the other techniques).
- Big concern: it is not fair to compare with Nie's method if this method has not been properly trained. So, include a fair comparison (for instance, no so many clouds) to then be technically able to claim, "compared with traditional target detection methods as well as deep learning algorithms".
Revise:
- the name of all section shall be in capital fonts, for instance,
3.2 candidate extraction results--> 3.2 Candidate extraction results
Revise them all.
Author Response
请参阅附件。谢谢你的建议。

Reviewer 3 Report
This article describes a new method for detecting ship targets under cloud interference and low-contrast ship targets under thin fog.
I find the paper can be interesting, but the contribution is not made clear enough in the Introduction part. In the introduction part, the authors say “We propose a new CFAR algorithm based on superpixel segmentation with feature points to solve the above problem”. The problems described are:
1) In the first case, the kind of clouds do not cover the target, the interference of small broken clouds.
- 2) In the second case the target is covered by thin clouds.
- 3) In the third case, the cloud's grayscale and the ship's grayscale values are close to each other and the coverage severity is relatively high.
I understand that this method improve the detection of ship in the first and second cases. The authors should be confirm this appreciation. If not, they should explain it better. It is not sufficiently detailed. The novelty, merit and/or contribution of this paper should be clearly shown in Introduction part.
I have some concerns.
In the Method Theory Explanation, they should be explain the characteristics (spatial and spectral resolution?) of MS and PAN images.
In line 186, they should be review or explain the expression diag (λ2……..).
In the Figure 4, they should review the images. For example, in the green zoom of (e) case, it seems that the geographic area does not have the same content.
In addition, the authors indicate in discussion part, the use of of GF-1 of visible panchromatic images with a spatial resolution of 2 meters. I understand that they use multispectral images and panchromatic images. They use pan-sharpening images?. GF-1 employs the CAST-2000 bus, it is configured with two 2 m Pan/8 m MS camera and a four 16 m MS medium-resolution and wide-field camera set.
In the comparison of experimental results, it seems that the proposed method obtains good results in case of the ship detection process, the results of the three indicators of accuracy, missed alarms rate and false alarms rate improve markedly with respect to other methods.
Author Response

(The authors gave the same response as above.)

Round 2
Reviewer 1 Report
The manuscript has been revised and improved by the authors, so I think it can be accepted for publication now.
Author Response
Thanks for your review and affirmation.
Reviewer 2 Report
All concerns were properly addressed by the Authors.
Author Response
Thanks for your review and affirmation.
This manuscript is a resubmission of an earlier submission. The following is a list of the peer review reports and author responses from that submission.